# Control of Fungal Diseases in Mushroom Crops while Dealing with Fungicide Resistance: A Review

**DOI:** 10.3390/microorganisms9030585

**Published:** 2021-03-12

**Authors:** Francisco J. Gea, María J. Navarro, Milagrosa Santos, Fernando Diánez, Jaime Carrasco

**Affiliations:** 1Centro de Investigación, Experimentación y Servicios del Champiñón, Quintanar del Rey, 16220 Cuenca, Spain; fjgea.cies@dipucuenca.es (F.J.G.); mjnavarro.cies@dipucuenca.es (M.J.N.); 2Departamento de Agronomía, Escuela Politécnica Superior, Universidad de Almería, 04120 Almería, Spain; msantos@ual.es (M.S.); fdianez@ual.es (F.D.); 3Technological Research Center of the Champiñón de La Rioja (CTICH), 26560 Autol, Spain; 4Department of Plant Sciences, University of Oxford, South Parks Road, Oxford OX1 2JD, UK

**Keywords:** cultivated mushrooms, dry bubble, cobweb, wet bubble, green mold, control, integrated disease management (IDM)

## Abstract

Mycoparasites cause heavy losses in commercial mushroom farms worldwide. The negative impact of fungal diseases such as dry bubble (*Lecanicillium fungicola*), cobweb (*Cladobotryum* spp.), wet bubble (*Mycogone perniciosa*), and green mold (*Trichoderma* spp.) constrains yield and harvest quality while reducing the cropping surface or damaging basidiomes. Currently, in order to fight fungal diseases, preventive measurements consist of applying intensive cleaning during cropping and by the end of the crop cycle, together with the application of selective active substances with proved fungicidal action. Notwithstanding the foregoing, the redundant application of the same fungicides has been conducted to the occurrence of resistant strains, hence, reviewing reported evidence of resistance occurrence and introducing unconventional treatments is worthy to pave the way towards the design of integrated disease management (IDM) programs. This work reviews aspects concerning chemical control, reduced sensitivity to fungicides, and additional control methods, including genomic resources for data mining, to cope with mycoparasites in the mushroom industry.

## 1. Introduction

Mushrooms are worldwide cultivated and consumed. In 2013, the global edible mushroom industry size was valued at 34 B € [1]. The cultivated mushroom industry, in line with the vegetable and fruit production industry, is subjected to increasing pressure for a change in the productive systems. A consumer-driven shift is claiming for healthier products with an environmentally respectful background to cut down dependence on chemical fungicides. Cultivated mushrooms are grown indoors under controlled environmental conditions that facilitate the implementation of integrated disease management programs, combining chemical fungicides, biocontrol alternatives, and correct agronomical management (the choice of casing and moisture level, the disinfection method, and management at the time of infection) to prevent outbreaks and disease dispersion [2,3,4].

Fungal diseases are among the most serious disorders of mushroom crops, damaging yield and mushroom quality [2,5]. The preventive use of phytosanitary products is routinely applied on top of the casing layer, although multiple evidence of resistance has been reported from the second half of the 20th century to different fungicide groups such as dithiocarbamates (multisite activity) [6], methylbenzimidazole carbamates (MBC) (targeting mitosis and cell division, inhibitor of spindle microtubules assembly) [7,8,9,10,11] or demethylation inhibitors (DMI-fungicides) (inhibitor of sterol biosynthesis in fungal cells) [12,13]. However, some alternative fungicides such as metrafenone (benzophenone) (whose mode of action targets cytoskeleton and motor protein; actin/myosin/fibrin function), introduced in the mushroom industry by the mid-2010s, showed promising results [14,15,16]. The continued use of the same active substances to fight fungal parasites of mushroom crops can prompt the onset of resistant strains [10,17]. Prior to allowing new fungicides, the effect on commercial host strains must be studied both in vitro and in vivo before approval to prevent crop damage [18]. Since both mycoparasite and host are fungi, proven selectivity is the key aspect for suitable fungicides, which restricts the emergence of alternative chemicals [19].

Although some evidence of a defensive response of mushrooms to the attack of fungal parasites has been studied, such as the noted increasing production of laccase to fight green mold [20], the immune response of mushrooms to mycoparasite action still remains broadly unresolved [5], which represents a shortcut to design resistant strains through breeding programs [21]. Likewise, mechanisms of host–microorganism interaction are still mostly unknown. However, different genomic resources, including the release of host [22] and some mycoparasite genomes [23,24,25,26], have been disclosed within recent years. The knowledge available can, for instance, serve as a tool for mapping quantitative trait loci (QTL) of mycoparasite-resistant genes in the mushroom host through data mining [27,28].

The present work reviews the four most harmful fungal diseases in cultivated mushrooms: dry bubble disease (DBD), cobweb, wet bubble disease (WBD), and green mold, including reported symptoms and causal agents, chemical control, and resistant evidence to eventually introduce proved alternative control methods in the form of bio-based product application and active biocontrol agents. Ultimately, recently released genomic resources are reviewed and discussed as a tool to design strategies for breeding programs to produce resistant strains. Besides, the authors compile a guide of good practices to cope with fungal diseases by means of intensive hygiene and crop management.

## 2. Dry Bubble (*Lecanicillium fungicola*)

### 2.1. Causative Agent and Symptoms of Disease

The causative agent of dry bubble (DBD) in mushroom crops [29] is associated with two varieties of the species *Lecanicillium fungicola,* var*. fungicola* and var*. aleophilum*, historically described as *Verticillium fungicola* [30]. Dry bubble is a ubiquitous fungal disease reported occurring in different countries cropping edible mushrooms [31,32,33,34], parasitizing various cultivated mushroom hosts such as *Pleurotus ostreatus*, *Agaricus bitorquis,* and *Agaricus bisporus* [29,35,36]. DBD accounts for an estimated 20% of button mushroom crop losses globally [37], which for the European sector represents losses of approximately 300 M€ annually. The ascomycete *L. fungicola* (Figure 1a) is an asexual organism that has been detected by metataxonomic analysis to be present in the compost and casing used in mushroom cultivation even in asymptomatic crops [32]. It is remarkable that symptoms of the disease have not been reported to occur in the colonized compost (Phase III) before applying the casing material on top. Therefore, first symptoms are reported to occur in the casing material, rarely prior to the harvest when the induction to fructification is promoted by means of environmental cues [38]. Primary infection drives to undifferentiated masses of infected host tissue known as bubbles. Likewise, other symptoms such as stipe blow-up (distorted and broken stipes of fully developed mushrooms) (Figure 1a), spotting, and sunken lesions in the caps generate serious yield losses [2].

The vegetative mycelium (net of hyphae colonizing the compost and the casing before fructification) has been proved resistant to *L. fungicola* [29]. The infected host tissue (reproductive morphological stage, mushroom tissue) is covered by the parasite which produces spherical masses (clusters) of conidia covered by sticky mucilage that favors the disease dispersion through watering, fly vectors (phorid and sciarid flies), workers, or dirty machinery [30,38]. The air powered through ventilation systems can also play an important role in disseminating DBD inside the mushroom farm or even transporting it to nearby installations. The main primary source of *L. fungicola* mentioned in mushroom crops is the casing material and, especially, peat [39]. Besides, infected mushrooms, phorid flies, and contaminated equipment can also play an important role as a primary source of infection [40].

### 2.2. Chemical Control and Resistance

The severity of the damage caused by DBD has led to applying several control measures against its causative agent. These preventive measures refer to the application of fungicides, control through the implementation of efficient cultural practices, sanitation, and attempts on the application of biological control.

In respect to the preventive fungicides applied to cope with DBD, the emergence of active substances from the group of methyl benzimidazole carbamates (MBC) in the late 1960s enabled efficient control of different fungal diseases affecting mushroom crops, including DBD as a major concern [41,42,43]. However, the continuous application of the same fungicides was quickly conducted to reduce sensitive and resistant *L. fungicola* strains, in addition to cross-resistance among MBC fungicides [7,8]. Bonnen and Hopkins [44], when evaluating fungicide response against isolates of *L. fungicola*, collected over a period of 45 yr, detected cross-resistance between benomyl and thiabendazole, negative cross-resistance between these two benzimidazole fungicides and the carbamate fungicide diethofencarb, and relatively high resistance towards chlorothalonil (chloronitrile, whose mechanism of action reduces and thereby deactivates glutathione). Evidence of resistance to the widely used benzimidazole and ergosterol demethylation inhibiting fungicides, including benomyl (ED_50_ = 415.45–748.12 mg L^−1^), carbendazim (ED_50_ = 1123.87–1879.59 mg L^−1^), and iprodione+carbendazim (ED_50_ = 415.45–748.12 mg L^−1^) have been also reported among Iranian strains isolated from diseased button mushrooms [31]. Besides, moderately reduced sensitivity to iprodione was noted among Serbian *L. fungicola* isolates [45], and resistant isolates to this active substance were also reported in Spain [46].

Prochloraz-Mn, currently approved imidazole fungicide to fight DBD in European mushroom crops, was introduced in 1983 to treat varieties of *Lecanicillium fungicola* resistant to benzimidazoles, WBD, and cobweb [47]. Likewise, evidence of reduced sensitivity to prochloraz-Mn with isolates showing ED_50_ values of 16.17 mg L^−1^ has been reported among Iranian strains [31]. In a major analysis of dose–response confronting *L. fungicola* to prochloraz-Mn, 105 isolates of *L. fungicola* from Spanish button mushroom crops collected between 1992–1999 were tested *in vitro*. *L. fungicola* sensitivity to prochloraz-Mn gradually diminished among newer isolates and moderately resistant strains were noted among the newest strains under study [13].

### 2.3. Alternative Control and Breeding

The number of biocontrol agents available to cope with mycoparasites of mushroom crops is still limited in the market [4,48]. However, currently, a fluent scientific activity aims to generate alternatives for the overused and, in some cases, relatively low-efficient chemical pesticides. Table 1 reviews alternative bio-based formulations reported within the last 15 years to fight mycoparasites (*L. fungicola, M. perniciosa, Cladobotryum* spp., and *Trichoderma* spp.) of cultivated mushrooms.

Among them, aqueous extracts obtain from diverse biological matrixes have proved efficient to treat DBD in crop trials and to avoid parasite germination and mycelial growth in vitro. Aqueous extracts obtained from composted bio-waste materials, known as compost teas (CT), exhibit antifungal properties to control plant pathogens [72]. Aerated and non-aerated compost teas obtained from spent mushroom compost (SMC), grape marc compost (GMC), crop residues compost (CRC), and crop residues vermicompost (CRV) provided significant in vitro suppression of *L. fungicola* [53]. In further work, the authors reported that the microbial community of these compost teas is the main factor responsible for the antagonistic effect against mycoparasites [73], which is in accordance with the state-of-the-art [74]. The selective suppressive effect noted in teas from SMC towards the mycoparasites could be related to the selectivity of mushroom compost for the growth of *A. bisporus*, which means that the native microbiota inhabiting this environmental niche is innocuous for the host while exhibiting fungitoxicity against the mycoparasites. After the evaluation of 16S rDNA-based denaturing gradient gel electrophoresis (DGGE) profiles (a technique used to separate DNA fragments by length based on their melting characteristics), they showed higher bacterial richness, diversity, and evenness values in aerated compared to non-aerated compost teas and noted high levels of siderophore (low-molecular-weight high-affinity iron-chelating compounds secreted by microorganisms that serve as iron carrier across cell reducing the availability of iron, an essential micronutrient for the mycoparasite) production in teas from CRC and CRV, high and consistent cellulase activity in teas from GMC and high protease activity in all the aerated compost teas under study, especially SMC [73]. Application of compost teas has been reported as innocuous for button mushroom crops since their application had no significant deleterious effect on the quantitative production parameters for the crop (yield, unitary weight, biological efficiency, and crop earliness) and compost teas incorporated into growth media in vitro was not fungitoxic for the host [75]. Therefore, the fungitoxic effect observed in vitro against *L. fungicola* and the results achieved showing a better response of aerated compost tea from SMC in comparison to prochloraz-Mn to prevent dry bubble disease in crop trials [54] suggest that the application of compost teas could be a suitable alternative to common fungicides. In addition to compost teas, EOs from aromatic plants have also shown fungitoxic effect in plate (cinnamon, thyme, and clove oils were the most effective in inhibiting the mycelial growth and conidia germination of *L. fungicola*) and effectiveness in crop applications (particularly effective in application after infestation) [56]. However, the selectivity of EOs towards the parasite is reduced and all the EOs tested by Geösel et al. [76] caused damage in the crop, inhibiting the growth of host mycelium and causing necrotic spots on mushroom caps. In this respect, selectivity trials must be conducted in addition to toxicity tests for the selection of more efficient products. Mehrparvar et al. [31] when evaluating the antifungal activity of eleven EOs against *L. fungicola* and *A. bisporus* noted different selectivity values (associated with the singular enzymatic pathways of the parasite and the host), highlighting the EOs of *Zataria multiflora* (main components thymol and carvacrol) and *Satureja hortensis* (main component carvacrol) as the most selective. 

The mechanism of host–pathogen interaction driving mycoparasitism of mushroom crops is still unknown, which is a limitation to develop breeding programs aimed to leverage host varieties resistant to fungal parasites. Zied et al. [77] reported different degrees of tolerance to the pathogen among 15 commercial strains of *A. bisporus* assayed in crop trials. The comparison of sensitive and resistant host strains can facilitate the identification of candidate genes involved in defensive response and also identify promoters suitable to ensure specificity of expression. The commercial white button mushroom line of *A. bisporus* has been proved to express specific genes differentially after infection [78]. DBD symptoms related to cap spotting can be a result of secondary metabolites produced by the parasite during conidia germination. The recently released genome of *L. fungicola* strain 150-1 (GenBank Ac. No. FWCC00000000) (Table 2) revealed 38 biosynthetic gene clusters for secondary metabolites including 8 PKSs (PolyKetide Synthases), 21 NRPS (Non-Ribosomal Peptide Synthetases) or NRPS-like clusters, 3 PKS-NRPS hybrids, 5 terpene synthases, and 1 indole cluster [23]. Further investigation and data mining can contribute to design programs for classical breeding and genetic modification with the aim of generating resistant varieties.

## 3. Cobweb Disease (*Cladobotryum* spp.)

### 3.1. Causative Agent and Symptoms of Disease

Cobweb is a globally widespread pathology whose causative agent corresponds with several Ascomycota species described as harmful in different mushroom-producing countries and all belonging to the genera *Cladobotryum* spp. Among them, *C. dendroides* [26], *C*. *mycophilum* [79], *C. protrusum* [25], or *C. varium* [80] have been described as parasites of cultivated mushrooms but also infecting mushrooms in the wild [81]. *Cladobotryum* spp. is a versatile parasite with a broad host range; among the reported cultivated species susceptible to cobweb are button mushrooms (*Agaricus* spp.), oyster mushrooms (*Pleurotus* spp.), shiitake (*Lentinula edodes*), beech mushroom (*Hypsizigus marmoreus*), winter mushroom (*Flammulina velutipes*) or reishi (*Ganoderma* spp.) [80,82,83,84,85,86,87]. The taxonomy and morphological features associated with the different parasites have been thoroughly described [88]. The prevalence of cobweb disease in commercial mushroom crops has been reported to vary between 6.8-33% in India, Turkey, or Spain [85]. Heavy losses up to 40% were noted when the disease reached epidemic proportions in the UK and Ireland in the mid-1990s [89].

Symptoms of the disease include the initial development of aerial and light-whitish mycelium over the casing layer and infected carpophores (Figure 1b) that quickly evolves due to massive sporulation to a dense white-floury mass that engulfs the surface of the casing layer and the surrounding carpophores; therefore, reducing the crop area available [86,87,88]. It is also worth noting that due to the nature of the dry spores generated by *Cladobotryum* (Figure 1b), they are easy to disturb from the patches of disease during cropping operations such as watering or picking, or even by low air currents in the growing facility [89]. When conidia land on mushroom caps, they generate spots that makes the crop unsalable, ending in significant losses [2,88].

Cobweb appears more often at the end of the crop cycle (although the earlier it appears, the more devastating it can be), commonly during autumn and winter cycles [86]. Under unfavorable conditions, particularly when the relative humidity (RH) is low, most *Cladobotryu*m spores do not survive for long periods. However, the fungus produces microsclerotia, resistant structures that can germinate even when they are stored at 0% RH [90]. High humidity conditions outside the cropping rooms will facilitate the survival of the pathogenic conidia and their dispersal throughout the production area [88,91].

It is also worth noting that due to the nature of the dry spores generated by *Cladobotryum* (Figure 1b), they are easy to disturb from the patches of disease during cropping operations such as watering or picking, or even by low air currents in the growing facility. A single conidium or mycelium debris mobilized from the initial infection can generate a new secondary outbreak because of their asexual nature. Since mushroom cultivation is produced in localized areas with a high density of growers, the initial source of the disease has been associated with external contamination of mushroom substrates through airborne conidia from nearby aged crops [88] or even specimens infected in the wild [92].

### 3.2. Chemical Control and Resistance

To prevent disease outbreaks, complex hygiene procedures are introduced during the transport of cultivation inputs (substrates, casing materials) and at the growing facilities as detailed in Section 6 [2,93]. However, still, many growers have no adequate means of controlling mushroom diseases such as cobweb and thus the risk of disease occurrence is extremely high. To tackle disease occurrence, chemical products are often applied as preventive treatments for extensive outbreaks. Despite different families of active substances have been historically applied including prochloraz-Mn (DMI-fungicides), metrafenone (benzophenone), chlorothalonil (chloronitrile), or thiabendazole (MBC-fungicide), in the last years the license of some broadly applied active substances are not renewed, for instance, recently the EU Member States voted to ban chlorothalonil [94], and the expected scenario suggests increasingly restrictive legislation. The scarce range of available products in addition to the continuous exposure to the same active substances has also favored the occurrence of resistance among *Cladobotryum* strains, which negatively affect the efficiency of treatments [82]. McKay et al. [95] reported the detection of a point mutation causing substitution from tyrosine to cysteine that removed an Acc I restriction site in the β-tubulin gene sequence of *C. dendroides* isolates resistant to MBC fungicides. Besides, *C. mycophilum* Type II isolate 192B1 was proved benzimidazole-resistant but sensitive to prochloraz-Mn, although prochloraz-Mn was unable to prevent spotting symptoms of cobweb disease [11]. The evidence reported suggests that it is necessary to count on alternative products to mitigate resistance occurrence, for example, the highly selective and effective metrafenone [15]. Nevertheless, fungicide discovery for mushroom crops is challenging and requires a priori knowledge about specificity for the relevant parasite while showing low toxicity for the host. Henceforth, it is difficult to expect the inclusion of novel fungicides to manage cobweb disease in mushroom crops in the short term.

### 3.3. Alternative Control and Breeding

Bioactive compounds extracted from different sources are interesting alternatives with proven antifungal activity tested as useful products to manage cobweb disease (Table 1). EOs from cinnamon, geranium, and spearmint completely inhibited the growth of *C. dendroides*, both by contact and the volatile phase were efficient, but low selectivity was reported, and the authors suggested the EOs can damage the host mycelium [76]. EOs from the Mediterranean aromatic plants *Thymus vulgaris* (main compounds p-cymene and thymol) and *Satureja montana* (main compounds carvacrol and p-cymene) were selective and efficient to inhibit mycelial growth of *C. mycophilum in vitro*, and reported significant control of cobweb disease, comparable to the fungicide treatment with metrafenone, when infected *A. bisporus* crops were treated with these EOs at 1%; although crop losses were noted in the first flush, suggesting some fungitoxic effect and crop delay [61]. The impact of cobweb disease in *Agaricus* and *Pleurotus* crops was also reduced significantly by the active component of the biofungicide Timorex 66 EC, tea tree oil; however, it showed significantly lower toxicity against *C. dendroides* mycelium than prochloraz-Mn in vitro [58,96].

Even though the mechanism underpinning cobweb disease of mushrooms is still unknown, genetic resources available for the researchers are gradually increasing, which can facilitate the development of studies on functional genomics fostering IDM strategies and facilitating programs for host breeding. For instance, the sequenced mitochondrial genomes of *Hypomyces aurantius* H.a0001 [97] and *C. mycophilum* MT108299 [98] have provided genetic resources for intra-species identification of the causal agent of cobweb disease, which represents significant information to design tailored control tools for a targeted species in local areas. The first complete genomes of the genus *Cladobotryum* were recently released (Table 2) [25,26]. The complete genome of *C. protrusum* strain CCMJ2080 (NCBI genome Acc. No RZGP00000000) showed that the long-interspersed element (LINE) detected in the genome (0.60%) could be related to the occurrence of resistance to DMI fungicides such as the commonly used prochloraz-Mn. The authors specifically suggest that fungicide resistance in *Cladobotryum* spp. maybe related to point mutations in one of the target genes (*BcSdhB*, *cox10*, *cytb*, *CYP51*, *DHFR*, *DHPS, FKS1*, *FUR1*, and *tub2*) [21]. In this respect, fungicides from benzophenone (like metrafenone), pyrimidinamines, and quinazoline groups pointing actin cytoskeleton-regulatory complex protein (PF12761), and NADH-ubiquinone oxidoreductase (PF12853) are suggested as efficient to control *C. protrusum*. It is noteworthy that the genome encodes genes from CAZymes, secondary metabolites, or cytochrome P450s, contributing to mycotrophic lifestyle [25]. The later release of the complete genome of *C. dendroides* strain CCMJ2808 (NCBI genome Acc. No WWCI01000000) showed that, among different mushroom mycoparasites, *Cladobotryum* genus was closer to *Mycogone* than to *Trichoderma* according to phenotypic evidence [26]. Pathogenicity-related genes were predicted and analyzed, detecting gene families in *C. dendroides* strongly associated with pathogenicity (such as CAZymes and fungal effectors), virulence (genes related to secondary metabolites production), and resistance (*C. dendroides* shared few unique orthologous gene families with other mushroom mycoparasites such as *C. protrusum* and *M. perniciosa*) [26].

## 4. Wet Bubble (Mycogone Perniciosa)

### 4.1. Causative Agent and Symptoms of Disease

Wet bubble disease (WBD), caused by the mycoparasite *M. perniciosa*, is a worldwide disease affecting commercial cultivation of white button mushroom (*A. bisporus*) and other cultivated fungi such as *Pleurotus citrinopileatus* but also found in the wild, parasitizing a range of basidiomycetes [2,63,99,100,101,102]. For many years, outbreaks of WBD have been sporadic, but it is recently reported that the presence of this mycoparasite is expanding, particularly in China, where WBD can cause yield losses of about 15–30% [103,104,105,106]. A taxonomic and morphological description of *M. perniciosa* (Figure 1c) can be found in several studies [2,101,102].

*M. perniciosa* affects the morphogenesis of *A. bisporus* fruit bodies but not the vegetative mycelium [102]. The easily recognized symptoms of WBD include the presence of masses of deformed tissue with no signs of differentiation into stipe or cap, which can reach 10 cm in diameter (Figure 1c). The wet bubbles are initially white, fluffy, and spongy before they acquire a brown color and decay. Later, they can secrete amber liquid drops containing bacteria and spores and eventually rot, releasing an unpleasant smell [2]. A cross-section of deformed sporophores shows black circular areas just beneath the upper layer. The main source of inoculum is the casing material, while compost is not cited as a major source [2]. Like *L. fungicola*, *M. perniciosa* can be spread by water splashing and operators (contaminated tools, hands, clothes, etc.). Air can also transport conidia.

### 4.2. Chemical Control and Resistance

WBD is managed by cultural practices, sanitation, and chemical fungicides. By the mid-1970s, the control of WBD relied on the use of benzimidazole fungicides, mainly benomyl, because this fungicide was toxic against a wide range of fungi but was not effective in inhibiting the growth of most Basidiomycetes [41,42]. When these fungicides were firstly used in the mushroom industry, they provided excellent control of *M. perniciosa*. Besides, isolates from Spain and Serbia have been proved highly sensitive to iprodione and prochloraz-Mn [64,100]. Recently in China, with the aim of screening suitable fungicides for the control of WBD, 26 fungicides from different FRAC groups (https://www.frac.info/ (accessed on 23 February 2021)) were tested in vitro against *M. perniciosa* and the host *A. bisporus*, while 13 selected fungicides were tested in mushroom-growing rooms infected with *M. perniciosa*. Among the fungicides assessed, prochloraz-Mn (DMI-fungicide) and thiabendazole (MBC-fungicide) are still useful to manage WBD, fludioxonil (DMI-fungicide), diniconazole (DMI-fungicide), fenbuconazole (DMI-fungicide), and imazalil (DMI-fungicide) and were also reported as suitable alternatives to prevent the occurrence of resistant strains [106]. However, extensive research must be performed before further approval of fungicides to treat WBD, evaluating the presence of residues after treatment as well as clarifying toxicity over the environment and human beings. To date, no solid evidence of resistance occurrence has been reported among *M. perniciosa* strains which are certainly remarkable in comparison to the evidence reported for other mycoparasites. This could be related to the nature of the organism and a limited ability to generate mutations driven toward fungicide resistance.

### 4.3. Alternative Control and Breeding

Some EOs from aromatic plants can suppress the growth of *M. perniciosa* in vitro (Table 1). EO from savory (*Satureja thymbra*) expressed better antifungal activity against *M. perniciosa* than *S. pomifera* oil, and those of oregano and geranium expressed the strongest antifungal activity against the mycopathogen; since EOs are considered nontoxic and easily biodegradable, their application is also recommended for disinfection of commercial casing soil with 2% oregano oil before application on germinated compost [107]. Among a wide range of forty EOs under study, Lemon verbena (*Lippia citriodora*), lemongrass (*Cymbopogon citratus*), and thyme (*Thymus vulgaris*) oils selectively inhibited the growth of *M. perniciosa*. Lemon verbena or thyme oils were efficient to prevent wet bubble when *M. perniciosa* was inoculated in the casing. Besides, in crop trial, the application of these oils, nerol and thymol (two major compounds of these substances), at a concentration of 40 μL L^−1^ showed a similar yield to those blocks treated with the fungicide prochloraz-Zn [65].

Highly WBD-resistant wild strains of button mushrooms from China have been reported, and the library of simple sequence repeat (SSR) markers generated during the study conforms a useful toolbox for the design of breeding programs to generate commercial WBD-resistant strains in addition to the precision mapping of QTL of WBD-resistant genes in *A. bisporus* [28]. Developing integrated programs for the successful management of WBD of button mushrooms requires knowledge from the genetic diversity and phenotypic virulence of *M. perniciosa*. In this sense, the amplified fragment length polymorphism (AFLP) analysis technique has been proved as efficient to distinguish the genetic variation among *M. perniciosa* isolates from China [105]. Ultimately, the targeting of genetic markers associated with disease incidence and parasite virulence will generate resources to breed *A. bisporus*. Genome resources recently generated in China through the release of the highly virulent strain of *Hypomyces perniciosus* HP10 (NCBI database under accession number: SPDT00000000) genome (Table 2) provided insights into the parasitic behavior of the organism, including the identification of significantly expanded protein families of transporters/carriers required for mycoparasitism and adaptation to rough environments (protein kinases, CAZymes, peptidases, cytochrome P450, and secondary metabolites) [24].

## 5. Green Mold (*Trichoderma* spp.)

### 5.1. Causative Agent and Symptoms of Disease

Green mold is a devastating disease for mushroom farmers in crops such as button mushroom, oyster mushroom, shiitake, winter mushroom, or milky mushroom (*Calocybe indica*) [108,109,110,111,112,113]. In December 2015, massive green mold epidemics caused by *Trichoderma aggressivum* f. *aggressivum* was reported to occur in Hungary, with nearly 100% crop loss in the infected button mushroom beds [114]. Symptoms of disease detected in compost and casing surface have been described as extensive sporulating green patches covering the substrates and generating brown spotting on mushroom caps [113] (Figure 1d). Several species belonging to the genus *Trichoderma* have been described as a causative agent for the disease including *T. aggressivum* [114], *T. citrinoviride* [109], *T. pleuroticola,* and *T. pleuroti* [112] or *T. harzianum* [111]. Different degree of virulence among biotypes is described, highlighting *T. aggressivum* f. *europaeum* and *T. aggressivum* f. *aggressivum* as the most dangerous parasites in button mushroom crops [113,115,116]. Other mushroom-substrate-inhabiting species such as *T. atroviride* are considered host competitors with reduced damage for the crop [2,116]. Morphologically the genus *Trichoderma* is characterized for producing heavy sporulation of small conidia, initially, white colonies turned to yellow and green and finally dark green depending on the species (Figure 1d). The different species from this genus are difficult to identify through taxonomy, so molecular identification is more often required [69,117].

The emergence of *Trichoderma* on the compost could be due to contamination during spawning. *Trichoderma* grows well on carbohydrates, and in this sense, seed grain is an important source of food and is very vulnerable. Once installed on the compost, the pathogen is able to colonize large areas since it is favored by the distribution of the seed grains in the compost mass [3].

Critical periods are the time of spawning and packaging of the compost when it is necessary to take extra precautions. The stage of colonization of the substrate is also critical, as there must be a good control of the temperature. The disease can be spread by general operations during cropping such as watering and by mean of pest flies acting as vectors for green mold [109,115].

### 5.2. Chemical Control and Resistance

Since green mold was considered historically a minor disease due to the relatively low impact when Phase II compost farming was conducted (which consist of the incubation of spawned compost in the growing facilities), the studies evaluating the efficiency of fungicides to prevent green mold is reduced.

The imidazole prochloraz-Mn has been proved effective against both *T. pleuroticola* and *T. pleuroti*, causal agents of green mold in *P. ostreatus*, both in in vitro and in crop trials when artificially infected [112]. *T. pleuroti* was more sensitive to the fungicide than *T. pleuroticola,* which points to the identification of the target species as an important intermediate step to optimize effective fungicide treatments in mushroom crops. Besides, the *T. aggressivum* f. *europaeum* strains T76, T77, and T85, isolated from green mold infecting compost in Serbia, when confronted to a range of fungicides, showed the largest sensitivity to chlorothalonil and carbendazim and were less susceptible to iprodione, some resistant to thiophanate-methyl, and resistant to trifloxystrobin [118].

### 5.3. Alternative Control and Breeding

Biochemical substances such as EOs from plants (Table 1) have shown strong activity against *Trichoderma aggressivum* f. *europaeum*, particularly basil and mint oils [119]. However, as noted before, some of these substances exhibit low selectivity and may damage the crop.

In addition, commercial biofungicides such as Serenade^®^ WP (Bayern, Germany), based on *Bacillus velezensis* QST713, and Ekstrasol F SC (BioGenesis d.o.o., Belgrade, Serbia), based on *Bacillus subtilis* Ch-13 1, have been noted effective to control green mold disease when *T. aggressivum* f. *europaeum* T77 was artificially inoculated in mushroom plots, showing a biological treatment comparable or even better than those plots treated with prochloraz-Mn [120]. Although mechanisms involving biocontrol of green mold from biofungicides have been barely described, according to Pandin et al. [121,122], the mechanisms acting in the antagonism of *T. aggressivum* by *B. velezensis* QST713 may be a combination of (i) competence in the environmental niche through the formation of competitive biofilm to exclude *T. aggressivum*, (ii) production and secretion of compounds with antifungal activity against the parasite, and (iii) release of signaling molecules, such as volatile organic compounds (VOCs) or the quorum-sensing-controlled processes, which can lead to defense response in *A. bisporus* or reduce parasite activity. The environmental niche where mushrooms develop and fructify is a highly dynamic and rich microbiome where multiple agents act and its configuration may have a direct impact on yield performance and disease occurrence [32,123,124]. Bacteria genera inhabiting casing material, including *Bacillus* and *Pseudomonas*, can be responsible for the casing fungistasis of mycoparasites observed in the absence of the host mycelium [125], however, the host mycelium modifies the microbiome structure breaking the fungistatic equilibrium and the native microbiota results inefficient to suppress mycoparasites under conditions of high disease pressure [32,48]. Bearing in mind the described behavior, selected microbiota isolated from mushroom substrates can be effectively reintroduced during cultivation as biological control agents (BCAs) to control green mold. For instance, the casing gram-negative bacteria *Pseudomonas putida* has been noted to promote the growth of harmful mycelium of *T. aggressivum,* and conversely casing inhabitants have been proved to reduce the growth of the parasitic mycelium but enhancing sporulation (*Pseudomonas tolaasii*) [113]. Among fifty bacterial strains isolated from mushroom compost, *Bacillus subtillis* B-38 inhibited *T. harzianum* T54 (48.08%) and *T. aggressivum* f. *europaeum* T77 (52.25%) mycelium growth in vitro. In plot trials, those plots artificially inoculated with the cited *Trichoderma* strains and treated with *Bacillus subtillis* B-38 presented significantly lower disease incidence comparing to the control and results for disease control and yield harvested were comparable to the plots treated with prochloraz-Mn [126]. *Bacillus amyloliquefaciens* B-129 (NCBI GenBank Ac No KT692726) isolated from Phase II compost and *B. amyloliquefaciens* B-268 (NCBI GenBank Ac No KT692735) isolated from casing soil have proved antagonistic activity against *T. aggressivum* f. *europaeum* T77, *T. harzianum* T54, *T. koningii* T39, and *T. atroviride* T33 in vitro, which indicates broad-spectrum activity among these gram-positive bacteria towards members of the *Trichoderma* genus [127].

Understanding genetic cues driving green mold disease can be crucial for the development of breeding strategies to product-resistant host strains. In this context, QTL in *A. bisporus* associated with *Trichoderma* lytic enzymes and metabolites have been described [27]. The authors suggest that the ability of *Agaricus* to resist or adapt to *Trichoderma* lytic enzymes and antifungal metabolites could be related to the fitness of the host strain, lower incidence and disease severity is associated with the prompt colonization of the compost by *A. bisporus* before *T. aggressivum* develops. Due to the evidence reported, it is highly advisable to prevent early contamination at the beginning of the crop cycle during the compost production and mycelium germination when strict hygiene measures must apply to prevent contamination. Resistance associated with laccase activity as a defense response of *A. bisporus* to *T. aggressivum* toxins has been also described, particularly associated with *A. bisporus lcc2* gene (laccase 2) [20]. Laccase production in *A. bisporus* increased in vitro in strains U1 and SB65 to confront toxic metabolites produced by *T. aggressivum*. The commercial brown strains SB65 showed enhanced defensive mechanisms, which can contribute to generating resistant strains against green mold through breeding programs. Besides, cell-wall-degrading enzymes encoded by three *T. aggressivum* genes: *prb1* (encoding a proteinase), *ech42* (encoding an endochitinase), and a *β-glucanase* gene, have been described and differences of *β-glucanase* transcription between the sensitive *A. bisporus* strain U1 (off-white hybrid) and the resistant *A. bisporus* strain SB65 (large brown strain) were noted [128]. In this framework, a strategy to design resistant strains for green mold disease may be the identification of genes that contribute to disease development because preventing the expression of targeted genes could facilitate the inhibition of the disease.

## 6. Recommended Hygienic Measures for the Control of Fungal Diseases

### 6.1. Hygienic Measures Common to the Four Diseases Described

To cope with fungal diseases, a number of management strategies can be designed to prevent disease outbreaks and avoid disease dispersion. The following are the most common general hygienic measures [2,3,129]: (1) Any agronomical activity which requires entering the growing facility should always be carried out from newer crops to the elder ones; (2) Disease vectors should be avoided by controlling fly and mite populations; (3) Batches of raw casing materials should not remain near crop facilities, otherwise, sealed spaces must be designed for casing storage to prevent spore contamination.; (4) Remove all affected mushrooms before applying agronomical actions such as harvesting or watering; (5) Disinfection of clothes, footwear and tools in critical areas is key action; (6) Boxes employed for mushroom collection must be disinfected before entering the growing facility and never come from nearby contaminated crops in the farm; (7) Do not lengthen unnecessarily the crop cycle (2–3 flushes maximum); (8) Once the crop cycle is terminated, when available, a cooking-out step on the crops must be applied using steam water to kill pathogens followed by thoroughly cleaning and disinfection of empty facilities.

### 6.2. Hygienic Measures Especially Recommended for the Control of Cobweb Disease

Particularly in the case of cobweb disease [88,89,130]: (1) Do not water or manipulate cropping areas that are affected by cobweb. It is recommended to place damp paper over the affected area (infected casing material or basidiomes) in such a way that the edges exceeded the pathogenic colony by several centimeters to prevent conidia release from sporulating patches; (2) Avoid long periods of high humidity on the carpophores and ensure a good rate of evaporation through the compost to prevent conidia germination on damp mushroom caps; (3) Prevent air ventilation and dispose slightly of suboptimal temperature and relative humidity to prevent disease dispersion and conidia germination when several patches of the disease have been noted.

### 6.3. Hygienic Measures Especially Recommended for the Control of Green Mould Disease

Especially in the case of green mold disorder, the most important measurements are [2,129,130]: (1) Ensure good mixing of base materials during Phase I of composting and avoid the appearance of anaerobiosis zones; (2) Ensure that the whole mass of compost receives good pasteurization. There should be a good flow of air through the whole mass to prevent the development of *Trichoderma* due to anaerobiosis; (3) Dispose of adequate air filters in Phase II rooms, areas of spawning, and incubation rooms; (4) Prevent excessive moisture in the final compost (<70%); (5) Control the levels of ammonia during phase II; (6) The packaging area must be sealed and equipped with positive pressure filtered air. Intensive cleaning of these areas must be performed at least once a week; (7) The spawning equipment must be disinfected at the end of the day; (8) Avoid unnecessary operations and temperatures of over 27–28 °C during the incubation phase.; (9) To prevent cap spotting, avoid humid conditions on the caps and poor evaporation rate through the carpophores when they develop.

## 7. Conclusions

The present work offers a comprehensive review in respect to the most harmful mycoparasites of cultivated mushrooms by describing the causal agents and disease symptoms of the worldwide occurring DBD, cobweb disease, WBD, and green mold and describing attempts for chemical control and evidence of disease resistance to finally introduce alternative methods for disease control and genomic resources to design programs for host breeding. This review is based on the authors’ long-lasting experience on the topic to compile topic knowledge as a toolbox for the design of strategies for integrated disease management of fungal diseases in mushroom crops, prevent the outbreak of resistant isolates, and eventually fight resistant variants through alternative control measurements.

In summary, evidence of reduced sensitivity and resistance occurrence has been reported among fungal parasitic strains continuously exposed to most of the active substances historically employed by the mushroom industry. Noteworthy, continuous exposure to the same active substances is related to a consistent trend, indicating higher tolerance among the most recent isolates. The great variability observed among fungicides with a different mode of action highlights the necessity for research studies to select suitable active substances that can be complementary. Complementary treatments may avoid overusing a single product to preclude the occurrence of resistant strains. Hence, growers and the mushroom industry have several challenges to overcome risks derived from the resistant evidence described, including the nature of the crop (increasing the pesticide doses is not a suitable alternative to fight mycoparasites), the restrictive legislative environment, and consumer demand for healthier and environmentally respectful cropping systems.

Natural plant-derived fungicides can provide a wide variety of compounds as an alternative to synthetic fungicides once tested selective for parasites and safe both for human health and the environment. Compost teas obtained from different agronomical wastes can be also efficient alternatives for the control of fungal diseases due to the intensive activity of the microbial community inhabiting these broths. Besides, comparing to the commercial biocontrol agents in the market, strains isolated from mushroom substrates may represent a sustainable solution tailor-made to this particular crop since the biocontrol agents will be perfectly adapted to the environmental niche associated with the mushrooms. Anyhow, the cost-effectiveness of treatments alternative to chemical fungicides must be taken into account and, in our opinion, both types of preventive treatments must cohabit in the short- and medium-term.

Genomic resources regarding mushroom mycoparasites have been generated within the last ten years. Further efforts must be taken into account to generate larger datasets through sequencing with the aim of enlightening the mechanism involved in host–parasite interactions, and ultimately design programs for the breeding of resistant varieties.

Intensive cleaning of compost yards and growing facilities, the integration of bioactive compounds, and the generation of further genomic data with a deepening understanding of available resources through data mining will be key inputs for designing IDM programs while integrating approved fungicides and good agricultural practices.

## Figures and Tables

**Figure 1 microorganisms-09-00585-f001:**
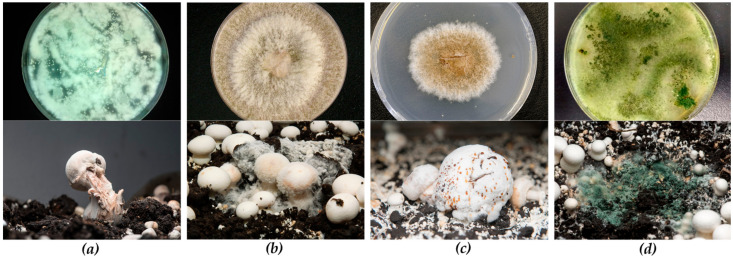
Mycoparasite plated on potato dextrose agar (PDA) (top pictures) and disease symptoms in *Agaricus bisporus* commercial crops (bottom pictures): (**a**) *Lecanicillium fungicola*/dry bubble disease (DBD); (**b**) *Cladobotryum* spp./cobweb disease; (**c**) *Mycogone perniciosa*/wet bubble disease (WBD); (**d**) *Trichoderma* spp./green mold disease.

**Table 1 microorganisms-09-00585-t001:** Alternative bio-based formulations employed to fight mycoparasites of cultivated mushrooms: EOs from botanicals, plant extracts, and compost teas (last 15 years).

Target Mycoparasite/Disease	Bio-Based Formulation	Biocontrol Activity	Proposed Mechanism	Reference
*Lecanicillium fungicola*/Dry Bubble Disease (DBD)	Three concentrations (5%, 10%, and 15%) filtered, microfiltered, and sterilized aerated compost teas (ACT) from grape marc compost.	Filtered and microfiltered ACT exhibited 100% inhibition of mycelium growth in vitro.	Compost excreted siderophores (due to the microorganisms present in grape marc compost) which were responsible for inhibiting the mycelium growth.	Dianez et al. [49]
	Essential oils (EOs) of *Matricaria chamommilla, Mentha piperita, M. spicata, Lavandula angustifolia, Ocimum basilicum, Thymus vulgaris, Origanum vulgare, Salvia officinalis, Citrus limon* and *C. aurantium*.	Oils from oregano and thymus species, containing phenolic compounds (carvacrol and thymol) showed the best inhibitory activity against the *L. fungicola *in vitro.	Components of EOs are responsible for the inhibitory effect (linalyl acetate, linalool, limonene, α-pinene, β-pinene, 1,8-cineole, camphor, carvacrol, thymol, and menthol).	Soković and van Griensven [50]
	EOs from lavender, anise, chamomile, fennel, geranium, oregano, parsley, and sage.	Oregano and geranium oils showed the most toxic effect against the *L. fungicola* var. *fungicola* when exposed to the volatile phase.	Components of EOs are related to their toxicity, oregano oil had a high content of carvacrol and thymol, geranium oil main components were citranelol and geraniol.	Tanović et al. [51]
	Untreated, autoclaved, or microfiltered non-aerated compost teas (NCT) from different sources (spent mushroom substrate, olive oil husk + cotton gin trash composted and mixed with rice husk, grape marc compost, and cork compost) were assayed against three *L.* fungicola isolates. Compost:water ratios of 1:4 and1:8 (w/v) were used and extraction periods of 1, 7, and 14 days.	Untreated NCT obtained the same % of mycelium inhibition as prochloraz-Mn. Autoclaving or microfiltration lost the activity. A period of 1–7 days and 1:4 dilution is recommended.	Antifungal activity due to the action of the active microbiota in the CT.	Gea et al. [52]
	ACT and NCT were obtained from four different composts: spent mushroom substrate compost, grape marc compost, greenhouse horticultural crop residues compost, and vermicompost.	ACT and NCT filtrates suppressed the mycelial growth of the mycopathogen in vitro. Sterilization by autoclaving or microfiltration removed partially or totally the inhibitory effect.	The efficacy of ACT and NCT depends on the microbiota present in them.	Marin et al. [53]
	NCT and ACT from SMS (spent mushroom substrate), one with mineral soil and the other with peat.	NCT and ACT from SMS significantly inhibited (100 %) the in vitro mycelial growth of *L. fungicola*. Treatments with aerated compost teas from SMS including peat-based casing reduced DBD incidence by 34–73 % in two crop trials, compared to an inoculated control.	Antifungal activity due to the action of the active microbiota in the CT.	Gea et al. [54]
	EOs from *Citrus limonum*, *Citrus aurantium*, *Zataria multiflora*, *Satureja hortensis*, *Mentha pulegium*, *Mentha piperita*, *Anethum graveolens*, *Foeniculum vulgare*, *Artemisia dracunculus*, *Artemisia sieberi* and *Pelargonium roseum*.	The EOs of thyme (*Z. multiflora*) and savory (*S. hortensis*) showed to be the most effective one since they showed the highest antifungal activity against mycoparasite and the best selectivity index between pathogen and host.	High antifungal activity of EOs thyme and savory due to the presence of phenolic components such as thymol and carvacrol detected by gas chromatography.	Mehrparvar et al. [55]
	EOs from *Melissa officinalis*, *Thymus vulgaris*, *Origanum vulgare*, *Eucalyptus globulus*, *Cinnamomum zeylanicum,* and *Syzygium aromaticum.*	Cinnamon and clove oils (0.4%) and thyme oil (0.8%) were the most efficient to inhibit the growth of pathogenic mycelium and prevent conidia germination *in vitro*. Thyme oil was effective to prevent DBD when applied in post-infection at 0.8%.	The occurrence of the disease is higher when the oils are applied pre-infestation, due to the volatility of the oils. Oil treatments prevent pathogenic conidia germination when applied post-infection probably due to the presence of phenolic compounds in their composition.	Dos Santos et al. [56]
	Inhibitory and fungicidal activity of two EOs, cinnamon (*Cinnamomum verum*) and clove (*Syzygium aromaticum*) tested by microdilution, macrodilution fumigant, and macrodilution contact method.	Clove oil showed the strongest activity than cinnamon against, showing the lowest minimum inhibitory concentration (MIC) in vitro against *L. fungicola*. Macrodilution fumigant method showed stronger antifungal effect than contact method.	Antifungal activity could be related to the presence of phenolic compounds within the EOs tested, such as eugenol, dominant phenolic compound in clove oil with proved strong antimicrobial activity.	Luković et al. [57]
*Cladobotryum* spp./Cobweb disease	EOs from lavender, anise, chamomile, fennel, geranium, oregano, parsley, and sage.	Oregano and geranium oils showed the most toxic effect against the *Cladobotryum* sp. when exposed to the volatile phase.	Components of EOs are related to their toxicity, oregano oil had high content of carvacrol and thymol, geranium oil main components were citranelol and geraniol.	Tanović et al. [51]
	Biofungicide: tea tree oil (Timorex 66 EC), based on the EO of *Melaleuca alternifolia*.	Tea tree oil was less toxic than prochloraz–manganese in vitro against *C. dendroides* isolates. Tea tree oil (drench application at 1%) applied in infected trials caused a significant reduction in cobweb disease and was not toxic against the crop (*A. bisporus*).	Most components of tea tree oil (highest antifungal activity due to the components: terpinen-4-ol, α-terpineol, linalool, α-pinene, and β-pinene) have activity against a range of fungi.	Potočnik et al. [58]
	Inhibitory and fungicidal activity of two EOs, cinnamon (*Cinnamomum verum*) and clove (*Syzygium aromaticum*) tested by microdilution, macrodilution fumigant, and macrodilution contact method.	Clove oil showed the strongest activity than cinnamon, showing the lowest minimum inhibitory concentration (MIC) in vitro agains*t C*. *dendroides*. Macrodilution fumigant method showed a stronger antifungal effect than the contact method.	Antifungal activity could be related to the presence of phenolic compounds within the EOs tested, such as eugenol, dominant phenolic compound in clove oil with proved strong antimicrobial activity.	Luković et al. [57]
	EOs extracted from 12 botanicals: *Syzygium aromaticum, Pelargonium graveolens, L. angustifolia, Cupresus sempervirens, M. piperita, Santolina chamaecyparissus, Citrus sinensis, Pogostemon patchouli, Thymus mastichina, Thymus vulgaris, Eucalyptus globulus,* and *R. officinalis.*	EOs obtained from clove, peppermint, patchouli, and rose geranium showed high antifungal activity in vitro against *C. mycophilum* with very low ED_50_ levels of 1.6, 7.4, 0.6, and 0.3%, respectively.	GC-MS showed eugenol in clove (86.38%), L-Menthol (41.97%) in peppermint, patchouli alcohol (33.40%) in patchouli, and citronellol (31.51%) in rose geranium as main components of these fungitoxic EOs.	Dianez et al. [59]
	Aqueous extracts from seven dried botanicals: mint leaves and stem (*Mentha longifolia*), garlic bulb (*Allium sativum*), turmeric rhizome (*Curcuma longa*), ginger rhizome (*Zingiber officinale*), clove seeds/buds (*Syzygium aromaticum*), cinnamon seeds (*Cinnamomum zeylanicum*), and neem leaves (*Azadirachta indica*).	*Syzygium aromaticum*, exhibited the maximum inhibition (99.48%) in vitro against *C. mycophilum* in amended PDA.	The spectra corresponding to the bioactive chemical constituents in *S. aromaticum* using Fourier transform infrared (FTIR) spectroscopy showed maximum intensive peak at 3375 cm–1 that represents the OH groups, this peak could correspond to antifungal phenolic compounds.	Idrees et al. [60]
	EOs obtained by hydrodistillation from five aromatic plants *(Lavandula × intermedia*, *Salvia lavandulifolia*, *Satureja montana*, *Thymus mastichina,* and *Thymus vulgaris*).	*T. vulgaris* and *S. montana* (ED_50_ = 35.5 and 42.8 mg L^−1^, respectively) showed the highest toxicity in vitro for inhibiting the mycelial growth of *C. mycophilum*, and the best selectivity between the pathogen and *A. bisporus.* EO from *T. vulgaris* showed some efficacy at controlling cobweb disease when used at the 1% rate in artificially infected crop trials.	The antimicrobial compounds carvacrol (17.22%) for *S. montana* and thymol (25.78%) for *T. vulgaris* were the most abundant phenolic compounds of these EOs, that also content contained a significant proportion of the biological precursors of the phenolic components p-cymene.	Gea et al. [61]
*Mycogone perniciosa/*Wet Bubble Disease (WBD)	EOs isolated from savory (*Satureja thymbra*) and sage (*Salvia pomifera* ssp. *calycina*).	*S. thymbra* EO showed better antifungal activity against *M. perniciosa* than *S. pomifera* oil in vitro by the microatmosphere method, minimal inhibitory quantity (MIQ), of 0.05 μL mL^−1^ and minimal fungicidal quantity (MFQ), of 0.25 μL mL^−1.^	The antifungal activity is related to the composition of the EOs as assessed by GC-MS: in *S. thymbra,* oils were γ-terpinene (23.2%) and carvacrol (48.5%), while in *S. pomifera* oil were α-thujone (20.4%) and β-thujone (36.1%).	Glamočlija et al. [62]
	EO from *Critmum maritimum* extracted from fresh plant material.	The essential oil of *C. maritimum* possessed antifungal activity in *vitro*, with MIQ = 1 μL disc-1 and MFQ = 20 μL disc^−1^ against *M. perniciosa*.	The chemical composition of *C. maritimum* EOs related to antifungal activity. With the two most abundant components, α-pinene (26.29%) and limonene (31.74%) showing strong antifungal activity.	Glamočlija et al. [63]
	EOs from lavender, anise, chamomile, fennel, geranium, oregano, parsley, and sage.	Oregano and geranium oils showed the most toxic effect against the *M. perniciosa* when exposed to the volatile phase.	Components of EOs are related to their toxicity, oregano oil had a high content of carvacrol and thymol, geranium oil’s main components were citranelol and geraniol.	Tanović et al. [51]
	EOs from seven botanicals.	*Thymus vulgaris* oil possessed the highest antifungal activity in vitro against *M. perniciosa* by microatmosphere method, with MIQ and MFQ of 0.02 μL mL^−1^ of air. *Pistacia terebinthus* showed the lowest antifungal effect, MIQ and MFQ of 0.16 and 0.65 μL mL^−1^ of air.	Oils from thymus species contain phenolic compounds (carvacrol and thymol) showing proved antimicrobial activity.	Potočnik et al. [64]
	Forty EOs, seven pure terpenoids, and one phenylpropanoid.	Lemon verbena (*Lippia citriodora*), lemongrass (*Cymbopogon citratus*) and thyme (*Thymus vulgaris*) oils substantially inhibited the growth of *M. perniciosa* in vitro, with the best selectivity between pathogen and host. Lemon verbena or thyme oils was able to control the development of WBD in casing-infected trial with innocuous effect to the host.	The main components of these oils, nerol and thymol, determined by GC–FID and GC–MS, showed antifungal activity, selective to the pathogen *M. perniciosa*.	Regnier and Combrinck [65]
	EOs from clove, castor, eucalyptus, olive, citrullina, and cinnamic aldehyde	Cinnamon oil in the form of cinnamic aldehyde (2.5 and 5 μL L^−1^) and eucalyptus oil (2.5 and 7.5 μL L^−1^) were the most effective to inhibit *M. perniciosa* in vitro with non-detrimental effect to the host strain at these doses.	Antifungal activity probably related to the chemical composition of the EOs and extracts (mainly due to the presence of phenolic compounds).	Sabharwal and Kapoor [66]
	Organic extracts of seeds of *Moringa peregrina.*	Moringa seed extracts inhibited the growth of both *M. perniciosa* in lower concentration than *A. bisporus*, through in vitro tests.	Candidates for the antifungal components in the seeds can be fatty acids like oleic acid and palmitic acid. Secondary metabolites such as sothiocyanates (99.9% of the volatile components of Iranian Moringa seeds) are also candidates for antifungal activity.	Shokouhi and Seifi [67]
*Trichoderma* spp./Green mold	Essential oils (EOs) of *Matricaria chamommilla, Mentha piperita, M. spicata, Lavandula angusti folia, Ocimum basilicum, Thymus vulgaris, Origanum vulgare, Salvia officinalis, Citrus limon* and *C. aurantium.*	Oils from oregano and thymus species, containing phenolic compounds (carvacrol and thymol) showed the best inhibitory activity against the *T. harzianum* in vitro.	Components of EOs are responsible for the inhibitory effect (linalyl acetate, linalool, limonene, α-pinene, β-pinene, 1,8-cineole, camphor, carvacrol, thymol, and menthol).	Soković and van Griensven [50]
	EO was *extracted from* Lippia alba by Clevenger hydrodistillation.	*L. alba* EO presented antifungal activity, with MIC of 0.6 mg mL-1 and MFC of 1.250 mg mL-1, against *Trichoderma viride*.	Geranial identified by GC-MS and NMR was described to be the main fungicidal component of this EO (50.9% of the chemical composition of *L.* alba EO)	Glamočlija et al. [68]
	Biofungicide: Timorex Gold (BM 608) EC (Stockton-Agrimor, Petach Tikva, Israel) based on tea tree oil (tea tree oil 23.8%; solvent 65.4%; ethanol 4.0%; NaOH 2.3%, and surfactant 4.5%), based on the EO of *Melaleuca alternifolia*.	Tea tree oil did not exhibit significant antifungal activity in vitro (ED_50_ = 11.9–370.8 mg L^−1^) against *T. atroviride, T. koningii, T. virens, T. aggressivum f. europaeum, and T. harzianum.* The biofungicide based on *B. subtilis* demonstrated greater effectiveness in preventing disease symptoms than tea tree oil. However, when combined with prochloraz-Mn, tea tree oil showed higher antagonism.	Most components of tea tree oil (highest antifungal activity due to the components: terpinen-4-ol, α-terpineol, linalool, α-pinene, and β-pinene) have activity against a range of fungi.	Kosanović et al. [69]
	EOs extracted from 12 botanicals: *Syzygium aromaticum, Pelargonium graveolens, L. angustifolia, Cupresus sempervirens, M. piperita, Santolina chamaecyparissus, Citrus sinensis, Pogostemon patchouli, Thymus mastichina, Thymus vulgaris, Eucalyptus globulus and R*. *officinalis.*	*Trichoderma aggressivum* f.sp. *europaeum* was less sensitive than *C. mycophilum* to the same EOs tested in vitro. It was most inhibited by peppermint, patchouli, and rosemary EOs with ED_50_ levels of 12.7%, 11.7%, and 3.4% in the growth medium, respectively.	GC-MS showed L-Menthol (41.97%) in peppermint, patchouli alcohol (33.40%) in patchouli, and citronellol (31.51%) in rose geranium as main components of these fungitoxic EOs.	Dianez et al. [59]
	An active film prepared by corn starch, polyvinyl alcohol, and carvacrol nanoemulsions (CNE).	More than 10% CNE applied into the film formulation showed inhibition against *Trichoderma* sp. Films with 25% CNE exhibited excellent antifungal activity with an inhibitory zone of 47 mm.	Efficient antifungal phenolic compound carvacrol.	Kong et al. [70]
	Plants essential oils and plant extracts of six medicinal plants (*Lippia* citriodora*, Ferula gummosa, Bunium persicum, Mentha piperita, Plantago* major, *and Salvadora* persica). In addition to a chimera peptide of camel lactoferrin (antimicrobial component of camel milk).	*L. citriodora, B. persicum,* and *M. piperita* treatments could completely prevent the growth of *T. harzianum* under in vitro conditions through disc diffusion method.	Antifungal activity probably related to the chemical composition of the EOs and extracts (mainly due to the presence of phenolic compounds).	Tanhaeian et al. [71]

**Table 2 microorganisms-09-00585-t002:** Genome features of host strain and mycoparasites were recently released as a source of information for breeding programs.

Strain/Disease	Genome (NCBI Acc. No)	Gene Prediction and Annotation	Lifestyle, Mycoparasitism, and Disease Resistance	Reference
*Agaricus bisporus* var. *bisporus* H97/Host	AEOK00000000	Wide repertoire of HTP*, β-etherases, multicopper oxidase, and CYP450* oxidoreductases up-regulated in mycelium-colonizing compost. The large gene of compost-induced CAZymes and oxidoreductases, together with high protein degradation and nitrogen-scavenging abilities.	Genetic and enzymatic mechanisms governing adaptation of *A. bisporus* to the selective substrate employed in mushroom cultivation and the fructification in the casing layer.	Morin et al. [22]
*Lecanicillium fungicola* strain 150-1/DBD *	FWCC00000000	37 biosynthetic gene clusters for secondary metabolites including 8 PKSs*, 21 NRPS* or NRPS-like clusters, 3 PKS-NRPS hybrids, 5 terpene synthases, and 1 indole cluster.	Analysis of these gene clusters is ongoing and could provide insight into the mechanistic of fungus–fungus interactions.	Banks et al. [23]
*Cladobotryum protrusum* strain CCMJ2080/Cobweb	RZGP00000000	The sequenced genome contained 412 CAZymes, 143 secondary metabolites, P450, and 1038 and 453 PHI (pathogen–host interaction) and DFVF* genes.	Arrays of genes that potentially produce bioactive secondary and stress response-related proteins could be associated with the mycoparasitic lifestyle. Long interspersed element (LINE) detected in the genome (0.60%) could be related to the occurrence of resistance to DMI such as the commonly used prochloraz-Mn.	Sossah et al. [25]
*Cladobotryum dendroides* strain CCMJ2807/Cobweb	WWCI01000000	The sequenced genome contained 327 CAZymes, 116 secondary metabolites. The authors identified 336 (3.52%), 175 (1.83%), and 48 genes encoding for cytochrome P450, 175 major facilitator superfamily (MFS) transporters (Pfam domain assignment), and 48 ATP-binding cassette (ABC) transporters.	Pathogenicity-related genes were predicted in *C. dendroides* strongly associated with pathogenicity (such as CAZymes and fungal effectors), virulence (genes related to secondary metabolites production), and resistance.	Xu et al. [26]
*Hypomyces perniciosus* strain HP10/WBD*	SPDT00000000	336 CAZymes analysis of six classes, 91 secondary metabolites gene clusters including T1PKS), non-ribosomal peptide synthetase hybrids, terpene synthases, and NRPS, and pathogenicity-related Genes (including 399 proteases and 125 cytochrome P450 or hydrophobins).	Resource data generated identified genes characterized to explain the basis of the mycoparasitic lifestyle in *H. perniciosus* as causal agent of wet bubble disease.	Li et al. [24]

* DBD: Dry bubble disease; WBD: Wet bubble disease; CYP450: Cytochrome P450 oxidoreductase; HTP: Heme-thiolate peroxidase; DFVF: Database of fungal virulence factors; PKSs: PolyKetide synthases; NRPS: Non-ribosomal peptide synthetases.

## Data Availability

Not applicable.

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
