# Peer review of "Control of Fungal Diseases in Mushroom Crops while Dealing with Fungicide Resistance: A Review"

_microorganisms, 2021, doi:10.3390/microorganisms9030585_

Round 1
Reviewer 1 Report
The theme discussed by authors is undoubtedly very important for the mushroom-producing industry, since the similar nature of mushrooms and their fungal parasites significantly limits the range of possible chemical protective agents. The language level is good and does not need any serious corrections. Authors collected a large volume of information concerning the fungal species causing the diseases, fungicides used for their control, as well as alternative biopreparations. Additionally, some data of the current genomic studies are mentioned, which probably provide a success in the clarification of the pathogen-host interactions and the search and selection of some mushroom lines, which would serve as a donors of resistance. Indeed, authors have a large experience in this field as they referenced a lot of their own publications. At the same time, the review in the current form has some drawbacks that should be addressed prior it can be accepted for publication. The most serious of them are the following:
First, authors declared four diseases to be described, but I did not find a subsection for the wet bubble disease. It should be corrected.
Second, the form of descriptions chosen by the authors considers a similar way of the information arrangement. This would provide a well structure of the review. However, there is a kind of difference in the description of control measures for different pathogens (I mean "Chemical control and resistance" sections for three described fungi). For example, subsection 2.2 describes chemical control (fungicides used for the pathogen and some data on resistance development) for the management of DBD. At the same time, the similar subsection 3.2 additionally describes sources of infection for the cobweb disease, ways of its spreading, and various sanitary measures required for this pathogen. In the case of the third pathogen, this section contains a very short information, but the description of ways of spreading present in the previous section (4.1); note that for the first pathogen no such information is included.
I consider information about the ways of spreading and infection sources should present for all pathogens; I would propose to place it at the end of sections 2.1, 3.1, and 4.1. I also consider that the review should contain information about agrotechnical and hygienic measures (like in the section 2.2) intended to prevent infection of mushroom crops. However, such information is generally similar for all mycoparasites included into the study, so probably it would be good to include it into a separate ("general") section of the review outside of the manuscript parts dedicated to each of the pathogen.
Third, subsection 3.4 and 4.4 (which have identical titles) seems to be not related with the general subject of the review. The review title is "Fungicide resistance in mushroom cultivation and management strategies: A review". The mentioned sections describe the possibility to use some pathogenic strains infecting mushrooms to control fungal diseases in plants. I doubt it is necessary to include this info in the manuscript.
Fourth, the main value in any review is the analytical part. Authors described a lot of information and compiled the main findings into tables, and this is good way for information presentation. At the same time, the chosen review title considers a kind of a concluding discussion about the resistance management strategies and promising directions in this field. The manuscript does not contain such discussion. In my opinion, authors could discuss the collected information: which fungicides are more preferable for use from the point of view of resistance development, which biopreparations seems to be a good alternative remedies with a high selectivity and have already been tested not only in vitro, but also under a small-scale (or maybe large-scale) industrial experiments. It would be also good to discuss the current achievements and directions in the field of antiresistant strategies in the industry and their prospects. For example, in plant protection, such antiresistant strategy usually include the alternation of chemical or biological preparations with different mechanism of action and different target sites to avoid the development of fungicide-resistant strains of pathogens. Authors mentioned about the necessity to avoid a long use of the same preparation, but did not clearly described possible alternatives (if exist) of the use of alternating treatments. Such kind of a concluding discussion would significantly improved the manuscript and made it much more valuable for readers.
Some minor remarks to the manuscript are listed below.
Introduction
A the end of thus section, authors write about the purpose of the study: a review of the current situation with four most harmful mushroom mycopathogens. I suggest it would be good to add some data on the size of losses caused by these pathogens to emphasize the scale of the problem.
Subsection 2.3.
Table 1. I suggest that the listed compounds may differ in their growth-inhibiting activity towards mushroom. In my opinion, it would be good to add such information (possible host growth inhibition or the lack of such effect) into the table, either as a separate column (say, just in a qualitative manner, not quantitative), or in the "Results" column to make it more clear. Maybe it would be good to organize the table in two parts, one (say, top) for neutral preparations and another (bottom) for preparations providing the negative effect on a host. I see some information of such kind in the text below the table (for example concerning compost teas), but it can be also mentioned in the table 1 to increase its value since the table contains a structured information about the preparations.
As alternative biopreparations, authors describe mainly essential oils and various substrate/compost or plant extracts. Table 1 mentions one peptide (the very end of the table), but no any results of its application are shown. Please, add them.
In crop protection, there are a lot of publications concerning the use of various bacterial or fungal metabolites to control fungal diseases. In many cases, even filtered culture broth obtained by submerged cultivation of some bacteria or fungi can be used to prevent the development of pathogenic fungi. Do you have any information about such studies in the field of mushroom protection?
Line 162-163. Did the authors of study mentioned at this place provide any explanation: why their compost tea preparation suppressed the growth of microscopic fungi, but did not significantly influences on mushrooms? If yes, then it would be good to add this info here.
Line 376: it is better to write gram negative instead of gram -. The similar remark is for the line 389.
Author Response
On behalf of the co-authors we really appreciate the efforts and valuable comments and suggestions provided by the reviewers. We have implemented major revisions to modify and improve the quality of the review. Please find below the list of changes implemented with the response to the reviewer suggestion marked in red. In order to facilitate the work of the reviewers we are submitting a new document without the control-change activated.
I hope the changes implemented satisfies the requirements of the journal to accept this Review.
Best regards,
Dr. Jaime Carrasco.
REVIEWER 1
The theme discussed by authors is undoubtedly very important for the mushroom-producing industry, since the similar nature of mushrooms and their fungal parasites significantly limits the range of possible chemical protective agents. The language level is good and does not need any serious corrections. Authors collected a large volume of information concerning the fungal species causing the diseases, fungicides used for their control, as well as alternative biopreparations. Additionally, some data of the current genomic studies are mentioned, which probably provide a success in the clarification of the pathogen-host interactions and the search and selection of some mushroom lines, which would serve as a donors of resistance. Indeed, authors have a large experience in this field as they referenced a lot of their own publications. At the same time, the review in the current form has some drawbacks that should be addressed prior it can be accepted for publication. The most serious of them are the following:
First, authors declared four diseases to be described, but I did not find a subsection for the wet bubble disease. It should be corrected.
Many thanks for this comments since we made a mistake and forgot to include the WBD by mistake. This section has now been included as “4. Wet bubble (Mycogone perniciosa)”.
Second, the form of descriptions chosen by the authors considers a similar way of the information arrangement. This would provide a well structure of the review. However, there is a kind of difference in the description of control measures for different pathogens (I mean "Chemical control and resistance" sections for three described fungi). For example, subsection 2.2 describes chemical control (fungicides used for the pathogen and some data on resistance development) for the management of DBD. At the same time, the similar subsection 3.2 additionally describes sources of infection for the cobweb disease, ways of its spreading, and various sanitary measures required for this pathogen. In the case of the third pathogen, this section contains a very short information, but the description of ways of spreading present in the previous section (4.1); note that for the first pathogen no such information is included.
These sections have been homogenized and relocated to improve text flow and ease reader comprehension as suggested.
Symptoms of the diseases together with evidences of main sources of disease inoculum and disease dispersion are respectively compiled in sections 2.1; 3.1; 4.1; 5.1. Subsections 2.2; 3.2; 4.2; 5.2 summarize chemical control of the diseases as referred by the literature and reported evidences of resistance. Section 5.2 contains less information due to the lack of evidences reported in the literature.
I consider information about the ways of spreading and infection sources should present for all pathogens; I would propose to place it at the end of sections 2.1, 3.1, and 4.1. I also consider that the review should contain information about agrotechnical and hygienic measures (like in the section 2.2) intended to prevent infection of mushroom crops. However, such information is generally similar for all mycoparasites included into the study, so probably it would be good to include it into a separate ("general") section of the review outside of the manuscript parts dedicated to each of the pathogen.
The section “6. Recommended hygienic measures for the control of fungal diseases” has been included to compile hygiene measurements to prevent and control fungal diseases.
Third, subsection 3.4 and 4.4 (which have identical titles) seems to be not related with the general subject of the review. The review title is "Fungicide resistance in mushroom cultivation and management strategies: A review". The mentioned sections describe the possibility to use some pathogenic strains infecting mushrooms to control fungal diseases in plants. I doubt it is necessary to include this info in the manuscript.
These two sections have been removed from the main text and any reference to these sections have been removed as well.
Fourth, the main value in any review is the analytical part. Authors described a lot of information and compiled the main findings into tables, and this is good way for information presentation. At the same time, the chosen review title considers a kind of a concluding discussion about the resistance management strategies and promising directions in this field. The manuscript does not contain such discussion. In my opinion, authors could discuss the collected information: which fungicides are more preferable for use from the point of view of resistance development, which biopreparations seems to be a good alternative remedies with a high selectivity and have already been tested not only in vitro, but also under a small-scale (or maybe large-scale) industrial experiments. It would be also good to discuss the current achievements and directions in the field of antiresistant strategies in the industry and their prospects. For example, in plant protection, such antiresistant strategy usually include the alternation of chemical or biological preparations with different mechanism of action and different target sites to avoid the development of fungicide-resistant strains of pathogens. Authors mentioned about the necessity to avoid a long use of the same preparation, but did not clearly described possible alternatives (if exist) of the use of alternating treatments. Such kind of a concluding discussion would significantly improved the manuscript and made it much more valuable for readers.
The section “7. Conclusions” has been rebuilt to include discussion as highlighted by the reviewer, in the same way the abstract has been also modified to include the approach.
Some minor remarks to the manuscript are listed below.
Introduction
A the end of thus section, authors write about the purpose of the study: a review of the current situation with four most harmful mushroom mycopathogens. I suggest it would be good to add some data on the size of losses caused by these pathogens to emphasize the scale of the problem.
This has been included in sections 2.1;3.1;4.1;5.1.
Subsection 2.3.
Table 1. I suggest that the listed compounds may differ in their growth-inhibiting activity towards mushroom. In my opinion, it would be good to add such information (possible host growth inhibition or the lack of such effect) into the table, either as a separate column (say, just in a qualitative manner, not quantitative), or in the "Results" column to make it more clear. Maybe it would be good to organize the table in two parts, one (say, top) for neutral preparations and another (bottom) for preparations providing the negative effect on a host. I see some information of such kind in the text below the table (for example concerning compost teas), but it can be also mentioned in the table 1 to increase its value since the table contains a structured information about the preparations.
As alternative biopreparations, authors describe mainly essential oils and various substrate/compost or plant extracts. Table 1 mentions one peptide (the very end of the table), but no any results of its application are shown. Please, add them.
In crop protection, there are a lot of publications concerning the use of various bacterial or fungal metabolites to control fungal diseases. In many cases, even filtered culture broth obtained by submerged cultivation of some bacteria or fungi can be used to prevent the development of pathogenic fungi. Do you have any information about such studies in the field of mushroom protection?
These products have been already reviewed: Carrasco, J.; Preston, G.M. Growing edible mushrooms: a conversation between bacteria and fungi. Environ. Microbiol. 2020, 22, 3, 858-872.
Section 5.3. include some of the information in respect to the work performed with Trichoderma, which is the mycoparasite more studied to be controlled by the application of biocontrol agents.
Line 162-163. Did the authors of study mentioned at this place provide any explanation: why their compost tea preparation suppressed the growth of microscopic fungi, but did not significantly influences on mushrooms? If yes, then it would be good to add this info here.
This is an aspect not described by the authors of this article. We have included the following statement in section 2.3: “The selective suppressive effect noted in teas from SMC towards the mycoparasites could be related to the selectivity of mushroom compost for the growth of A. bisporus, which means that the native microbiota inhabiting this environmental niche is innocuous for the host while exhibiting fungi toxicity against the mycoparasites.
Line 376: it is better to write gram negative instead of gram -. The similar remark is for the line 389.
Done
Reviewer 2 Report
Please see attached file.

Author Response
On behalf of the co-authors we really appreciate the efforts and valuable comments and suggestions provided by the reviewers. We have implemented major revisions to modify and improve the quality of the review. Please find below the list of changes implemented with the response to the reviewer suggestion marked in red. In order to facilitate the work of the reviewers we are submitting a new document without the control-change activated.
I hope the changes implemented satisfies the requirements of the journal to accept this Review.
Best regards,
Dr. Jaime Carrasco.
REVIEWER 2
General:
The authors review an array of mycoparasitic diseases of cultivated mushrooms by exploring the diseases and current treatment options. They also draw upon recent bioinformatic advances (e.g., genomic sequencing and resources) to shed light on new approaches for therapeutic design. I enjoyed reading this article and found it to be well written and informative. More background on the diseases and hosts, including global distributions, prevalence, etc. would be beneficial.
Global distribution and impact on yield (production losses) have been included.
Specific:
Line 39: Suggest including target or mode of action for each of the fungicides listed. This may be elaborated on in the subsequent chemical control sections.
The mode of action of the fungicide families employed for the comtrol of mycoparasites has been included in the Introduction section.
Line 40: Define DMI
Done- Introduction section.
Line 53: minor grammatical corrections needed for readability
Done.
Line 65: Italics
Done.
Line 71: use full scientific name if first introduction of the hosts
Done.
Where are the cultivated mushrooms typically grown (i.e., global locations)?
The following paragraph has been included in the introduction: “Cultivated mushrooms are worldwide cultivated and consumed. In 2013, the global edible mushroom industry size was valued at 34 B €”.
For information about the causative agent, suggest including global distributions and prevalence. Applicable for each disease and host.
Aspects concerning prevalence and losses reported in the literature have been included in paragraph 2.1;3.1;4.1;5.1.
Line 152: DGGE previously defined? Very long sentence.
Defined: “After the evaluation of 16S rDNA based denaturing gradient gel electrophoresis (DGGE) DGGE profiles (a technique used to separate DNA fragments by length based on their melting characteristics),…”
Line 191: PKs, NRPS previously defined?
Defined when they firstly appear. “including 8 PKSs (PolyKetide Synthases), 21 NRPS (Non Ribosomal Peptide Synthetases)”
Line 202, 308: write out full scientific names if not yet introduced.
Done when they firstly appear.
Line 305: Essential oils introduced previously as EO, although, I’m not sure the abbreviation is helpful or necessary.
Done. This is a redundant term throughout the article, therefore Essential oils are defined as EOs when they firstly appear, then EO/EOs is employed to refer essential oil/essential oils respectively.
Line 376, 389: Gram-negative, Gram-positive
Done.
Line 396: QTL previously used
Done.
Throughout the text, recommendation or insights are relatively limited. Comments about the potential for future discoveries based on genome sequencing and tools is evident; however, more commentary on the direction of alternative bio-treatments, connections between mycoparasitic distribution or severity and global issues (e.g., climate change) would be enriching.
Climate change should not be entirely related to the impact of fungal diseases. As noted this is a crop that is cultivated indoors, therefore relative humidity or external temperature is not related to the impact of the disease. This already discuss in the article.
Correction of minor grammatical errors and changes to sentence structure will support better readability.
Done
Fig. 1: very nice images
Table 1: helpful and informative but since the main text has only covered DBD, perhaps the table could be separated into each disease and presented along with the respective disease sub-section. Currently, a lot of information is presented but we have not read context and information about the other diseases (e.g., green mould) to this point. If the table is split into each disease, recommend adding sub-headings and grouping treatments accordingly (e.g., essential oils) so the reader can easily see the overall approach of the alternative option and then read about specifics amongst comparables.
This Table have been rebuilt following reviewers comments.
Table 2: Similar to table 1. Species that are relevant to DBD should be covered in this table as the other diseases have not yet been described in the Review. Suggest consistency in referencing within tables (i.e., number vs. name).
The species reported in Table 2 have been described in the review as causative agents of DBD cobweb and WBD.
Reviewer 3 Report
General:
The authors review an array of mycoparasitic diseases of cultivated mushrooms by exploring the diseases and current treatment options. They also draw upon recent bioinformatic advances (e.g., genomic sequencing and resources) to shed light on new approaches for therapeutic design. I enjoyed reading this article and found it to be well written and informative. More background on the diseases and hosts, including global distributions, prevalence, etc. would be beneficial.
Specific:
Line 39: Suggest including target or mode of action for each of the fungicides listed. This may be elaborated on in the subsequent chemical control sections.
Line 40: Define DMI
Line 53: minor grammatical corrections needed for readability
Line 65: Italics
Line 71: use full scientific name if first introduction of the hosts
Where are the cultivated mushrooms typically grown (i.e., global locations)?
For information about the causative agent, suggest including global distributions and prevalence. Applicable for each disease and host.
Line 152: DGGE previously defined? Very long sentence.
Line 191: PKs, NRPS previously defined?
Line 202, 308: write out full scientific names if not yet introduced.
Line 305: Essential oils introduced previously as EO, although, I’m not sure the abbreviation is helpful or necessary.
Line 376, 389: Gram-negative, Gram-positive
Line 396: QTL previously used
Throughout the text, recommendation or insights are relatively limited. Comments about the potential for future discoveries based on genome sequencing and tools is evident; however, more commentary on the direction of alternative bio-treatments, connections between mycoparasitic distribution or severity and global issues (e.g., climate change) would be enriching.
Correction of minor grammatical errors and changes to sentence structure will support better readability.
Fig. 1: very nice images
Table 1: helpful and informative but since the main text has only covered DBD, perhaps the table could be separated into each disease and presented along with the respective disease sub-section. Currently, a lot of information is presented but we have not read context and information about the other diseases (e.g., green mould) to this point. If the table is split into each disease, recommend adding sub-headings and grouping treatments accordingly (e.g., essential oils) so the reader can easily see the overall approach of the alternative option and then read about specifics amongst comparables.
Table 2: Similar to table 1. Species that are relevant to DBD should be covered in this table as the other diseases have not yet been described in the Review. Suggest consistency in referencing within tables (i.e., number vs. name).
Author Response
On behalf of the co-authors we really appreciate the efforts and valuable comments and suggestions provided by the reviewers. We have implemented major revisions to modify and improve the quality of the review. Please find below the list of changes implemented with the response to the reviewer suggestion marked in red. In order to facilitate the work of the reviewers we are submitting a new document without the control-change activated.
I hope the changes implemented satisfies the requirements of the journal to accept this Review.
Best regards,
Dr. Jaime Carrasco.
REVIEWER 3
M/S ref.: microorganisms -1124733
Title: Fungicide resistance in mushroom cultivation and management strategies: A review
Gea et al. described a review-article dealing with the ‘chemical control, reduced sensitivity to fungicides, integrated disease management (IDM) programs and alternative control methods’ in mushroom cultivation. The overall interest of the work is certainly very high. As also stated in the m/s, ‘fungal diseases are among the most serious disorders of mushroom crops damaging yield and mushroom quality’ and their control is necessary for achieving high performances in mushroom industry. Likewise, the presentation and the arrangement of the m/s are quite correct. Chemical and biological control are discussed, summarized and reviewed in the presently submitted work, along with the mushrooms’ resistance on fungicides. Under this approach, the submitted review-article presents interest. On the other hand, some of the first in the literature relevant papers are not mentioned in the long catalog of references and should be added:
Van Zaayen, A., 1982. Mushroom J. 113, 149–161.
Fletcher, J.T., Hims, M.J., Hall, R.J., 1983. Plant Pathol. 32, 123–131.
Challen, M.P., Elliot, T.J., 1985. Mycopathologia 90, 161–164.
Diamantopoulou et al. 2006, Sc. Hort 109, 190-95;
Chrysayi-Tokousbalides et al. 2007 Journal of Environmental Science and Health Part B, 41:571–583, 2006.
Chrysayi-Tokousbalides Crop Protection 26 (2007) 469–475
Kastanias et al. Bull. Environ. Contam. Toxicol. (2006) 77:149–154
Although the literature revision included is already quite extensive, some further references have been included as suggested by the reviewer:
Van Zaayen, A.; Van Adrichem, J. C. J. Prochloraz for control of fungal pathogens of cultivated mushrooms. Neth. J. Plant Pathol. 1982, 88, 203-213.
Fletcher, J. T.; Hims, M. J.; Hall, R. J. The control of bubble diseases and cobweb disease of mushrooms with prochloraz. Plant Pathol. 1983, 32, 123-131.
Diamantopoulou, P.; Philippoussis, A.; Kastanias, M.; Flouri, F.; Chrysayi-Tokousbalides, M. Effect of famoxadone, tebuconazole and trifloxystrobin on Agaricus bisporus productivity and quality. Sci. Hortic. 2006, 109, 190-195.
Chrysayi-Tokousbalides, M.; Kastanias, M.A.; Philippoussis, A.; Diamantopoulou, P. Selective fungitoxicity of famoxadone, tebuconazole and trifloxystrobin between Verticillium fungicola and Agaricus bisporus. Crop Prot. 2007, 26, 469-475.
Additionally, there is a serious problem on the usage of the English language; the abstract should be re-written and the whole text should be grammatically corrected.
The abstract and the rest of the text have been reviewed and certain paragraphs rebuilt to facilitate comprehension and improve text quality.
Decision: The review-article seems critical and correctly written. Minor revision of the basis of the referee’s comments is requested for the current submission.
Round 2
Reviewer 1 Report
Authors answered my comments and added the lost part of the review, as well as re-arranged the manuscript that significantly improved it. Now I see only some misprints and language errors that should be corrected prior publication (see, for example, some of them below). Thus, I consider the manuscript can be accepted for publication after a minor revision and language checking across the text.
Line 36: fibrin
Line 37: continued use (not used)
Table 1:
- please, look for misprints and unnecessary symbols (for example, concentra-tions in the first row/second column
- Proposed mecHanism (see column titles)
Table 2, column titles: genE instead of gen
The row of this table for the C. dendroides CCMJ2807: please, check the column 3 (The genome of contained...), the genome of which organism?
The same column, but the next row: secondaRy instead of secondaTy.
Line 176-177: "up to" is doubled.
Line 389: usually, numbers below 10 are typed by letters, so I would recommend to replace "encoded by 3..." with "encoded by three...".
Line 392: a strategy (not "an").
Section 6: since this is a review, then it would be good to give references for these recommendations for the sections 6.2 and 6.3 (if any, of course).
Author Response
Dear Reviewer,
On behalf of the co-authors we really appreciate the opportunity to implement minor revision in this review. Thanks for the efforts pointing out corrections and suggestions. All the changes suggested have been implemented. In addition we have made an effort to re-phrase some paragraphs and correct minor spelling mistakes in the Tables and throughout the text. Please find attached the cover letter including changes in reply to the comments (reply marked in red).
Best regards,
Jaime Carrasco, PhD
